# Inorganic Polyphosphate Promotes Colorectal Cancer Growth via TRPM8 Receptor Signaling Pathway

**DOI:** 10.3390/cancers16193326

**Published:** 2024-09-28

**Authors:** Valentina Arrè, Francesco Balestra, Rosanna Scialpi, Francesco Dituri, Rossella Donghia, Sergio Coletta, Dolores Stabile, Antonia Bianco, Leonardo Vincenti, Salvatore Fedele, Chen Shen, Giuseppe Pettinato, Maria Principia Scavo, Gianluigi Giannelli, Roberto Negro

**Affiliations:** 1Personalized Medicine Laboratory, National Institute of Gastroenterology “S. de Bellis”, IRCCS Research Hospital, Via Turi 27, Castellana Grotte, 70013 Bari, Italy; francesco.balestra@irccsdebellis.it (F.B.); rosanna.scialpi@irccsdebellis.it (R.S.); francesco.dituri@irccsdebellis.it (F.D.); maria.scavo@irccsdebellis.it (M.P.S.); 2Data Science, National Institute of Gastroenterology “S. de Bellis”, IRCCS Research Hospital, Via Turi 27, Castellana Grotte, 70013 Bari, Italy; rossella.donghia@irccsdebellis.it; 3Core Facility Biobank, National Institute of Gastroenterology “S. de Bellis”, IRCCS Research Hospital, Via Turi 27, Castellana Grotte, 70013 Bari, Italy; sergio.coletta@irccsdebellis.it (S.C.); dolores.stabile@irccsdebellis.it (D.S.); antonia.bianco@irccsdebellis.it (A.B.); 4Unit of Surgery, Department of Surgery Sciences, National Institute of Gastroenterology “S. de Bellis”, IRCCS Research Hospital, Via Turi 27, Castellana Grotte, 70013 Bari, Italy; leonardo.vincenti@irccsdebellis.it (L.V.); salvatore.fedele@irccsdebellis.it (S.F.); 5Division of Infectious Diseases, Washington University School in Medicine in St. Louis, 660 S Euclid Ave., St. Louis, MO 63110, USA; shenc@wustl.edu; 6Division of Gastroenterology, Department of Medicine, Beth Israel Deaconess Medical Center, Harvard Medical School, 330 Brookline Avenue, Boston, MA 02215, USA; gpettina@bidmc.harvard.edu; 7Scientific Direction, National Institute of Gastroenterology “S. de Bellis”, IRCCS Research Hospital, Via Turi 27, Castellana Grotte, 70013 Bari, Italy; gianluigi.giannelli@irccsdebellis.it

**Keywords:** colorectal cancer, inorganic polyphosphate, TRPM8 receptor, organoids, *ccnb1*, proliferation

## Abstract

**Simple Summary:**

Inorganic polyphosphate, a molecule composed of a few to several hundred orthophosphates, is involved in disparate pathological processes, including cancer development and progression. In this study, by analyzing biopsies derived from 50 subjects carrying colorectal cancer, collected by the Histopathology Unit of IRCCS “S. de Bellis,” we showed a significant discrepancy in the level of inorganic polyphosphate between tumoral and peritumoral counterparts, with the former displaying a higher concentration. Moreover, by employing in vitro and ex vivo approaches, we revealed the involvement of inorganic polyphosphate in colorectal cancer cell proliferation. Additionally, we identified the calcium channel TRPM8 as the inorganic polyphosphate receptor responsible for propagating the signal downstream and, ultimately, enhancing the expression of proliferative markers. Thus, our results recognize inorganic polyphosphate as a novel pivotal biomarker linked with colorectal cancer growth.

**Abstract:**

Background: Colorectal cancer (CRC) is characterized by a pro-inflammatory microenvironment and features high-energy-supply molecules that assure tumor growth. A still less studied macromolecule is inorganic polyphosphate (iPolyP), a high-energy linear polymer that is ubiquitous in all forms of life. Made up of hundreds of repeated orthophosphate units, iPolyP is essential for a wide variety of functions in mammalian cells, including the regulation of proliferative signaling pathways. Some evidence has suggested its involvement in carcinogenesis, although more studies need to be pursued. Moreover, iPolyP regulates several homeostatic processes in animals, spanning from energy metabolism to blood coagulation and tissue regeneration. Results: In this study, we tested the role of iPolyP on CRC proliferation, using in vitro and ex vivo approaches, in order to evaluate its effect on tumor growth. We found that iPolyP is significantly increased in tumor tissues, derived from affected individuals enrolled in this study, compared to the corresponding peritumoral counterparts. In addition, iPolyP signaling occurs through the TRPM8 receptor, a well-characterized Na^+^ and Ca^2+^ ion channel often overexpressed in CRC and linked with poor prognosis, thus promoting CRC cell proliferation. The pharmacological inhibition of TRPM8 or RNA interference experiments performed in established CRC cell lines, such as Caco-2 and SW620, showed that the involvement of TRPM8 is essential, greater than that of the other two known iPolyP receptors, P2Y1 and RAGE. The presence of iPolyP drives cancer cells towards the mitotic phase of the cell cycle by enhancing the expression of *ccnb1*, which encodes the Cyclin B protein. In vitro 2D and 3D data reflected the ex vivo results, obtained by the generation of CRC-derived organoids, which increased in size. Conclusions: These results indicate that iPolyP may be considered a novel and unexpected early biomarker supporting colorectal cancer cell proliferation.

## 1. Introduction

Colorectal cancer (CRC) remains the third most deadly cancer diagnosed worldwide and shows an increasing incidence in both developing and developed countries [1]. Patients with CRC exhibit a wide range of genetic and epigenetic alterations [2,3]. Sporadic or induced mutations in the CRC scenario often fall within a discrete set of tumor suppressor genes, like adenomatous polyposis coli (*APC*), SMAD4 family member (*SMAD4*), or those traditionally involved in the regulation of cell proliferation and the cell cycle, like *p53* [4]. In addition, mutations in oncogenes, particularly concerning genes such as Kristen rat sarcoma virus (*KRAS*), phosphatidylinositol 4,5-bisphosphate 3-kinase catalytic subunit-α (*PIK3CA*), or b-raf proto-oncogene (*BRAF*), confer hyperproliferative traits to intestinal epithelial cells that promote a genetic hypermutability scenario [5]. In addition to the unfavorable genetic predisposition, CRC development embraces an heterogenicity of factors spanning from lifestyle to environmental mutagens or, more recently, dysbiotic metabolites [6], although mechanistically, a number of queries remain unaddressed and novel potential candidates need to be disclosed. The complexity of CRC relies on the existence of several different molecular subtypes, characterized by apparently unrelated pathways of development. The major issues are, in fact, represented by the presence of multiple starting and diverging points during the steps from polyp to adenoma to adenocarcinoma [7]. Gaps in the current understanding of CRC onset include, for instance, the molecular lifestyle and environmental drivers of mutations that cause polyp formation, what background determines the switch to a malignant phenotype, and not least, what makes a CRC subtype more responsive to specific therapies than others [8]. The food additive inorganic polyphosphate (iPolyP) is a biological polymer structurally composed from 3 to over 1000 orthophosphates linked together by ATP-like bonds [9], and it can be enzymatically produced by living organisms, including bacteria and eukaryotes, or result as a product of inorganic catalysis [10]. While the synthesis and degradation of iPolyP in bacterial cells has been well-studied, the corresponding metabolic pathways in the eukaryotic counterpart are still poorly understood [11]. Mammalian iPolyP has been found in nuclei, mitochondria, lysosomes, platelets dense granules, and mast cells- and basophils-derived granules [12,13,14,15]. The iPolyP architecture suggests its role in energy metabolism; it is, in fact, nowadays considered a phosphate donor for ATP synthesis [16]. Thus, being considered an optimal source of energy, emerging literature has started to associate iPolyP with various pathophysiological processes over the past few decades, including inflammation-driven diseases [17,18,19,20,21,22], tumorigenesis [11], tumor metastasis [23,24], and cellular proliferation [25,26]. It has been shown that iPolyP mediates cellular proliferation by selectively upregulating mechanistic target of rapamycin (mTOR) phosphorylation at Ser2481 [25]. Three binding receptors have been identified for iPolyP, namely advanced glycosylation end-product specific receptor (RAGE), purinergic receptor P2Y1, and transient receptor potential cation channel subfamily M (melastatin) member 8 (TRPM8) [27,28], displaying entirely different transduction pathways, although all three have been linked to CRC progression and development. In particular, the RAGE receptor, known to bind to the so-called advanced glycation end-products (AGEs), signals to nuclear factor kappa B (NF-κB) and controls the expression of several genes involved in inflammation, which might foster CRC development [29]. Preliminary evidence revealed a novel pathway involving iPolyP/the RAGE receptor linked to the Wingless-related integration site (Wnt)/β-catenin signaling axis, of crucial importance for the regulation of major pathophysiological processes in tumor cells [30]. The purinergic receptor P2Y1, which binds to extracellular ATP, is markedly overexpressed in CRC in respect to the normal counterpart [31], although mechanistic studies need to be performed. Parallelly, recent literature has reported an overexpression of the TRPM8 receptor in CRC specimens, which correlates with poorer survival [32]. TRPM8 belongs to the transient receptor potential (TRP) cation channel superfamily, subfamily melastatin (M), member 8 (TRPM8), also known as the cold and menthol receptor 1 (CMR1) [33]. However, apart from slight evidence about iPolyP-mediated cell proliferation, not many studies have yet reported associations between iPolyP, its receptor, and cancer [34], perhaps due to difficulties in iPolyP quantification and the lack of comparative data for neoplastic and corresponding normal counterpart. The aim of our study was firstly to investigate whether iPolyP levels were enhanced in tumoral CRC tissue, compared with non-neoplastic counterparts of the same subject. Additionally, we wanted to rule out its role in the development and progression of the disease, as well as the receptor involved in iPolyP signaling, through in vitro and ex vivo strategies. Thus, we tested the amount of iPolyP in the tumoral and peritumoral tissues of enrolled colorectal cancer subjects, showing a significant discrepancy. Moreover, we determined the role of the TRPM8 receptor as a mediator of iPolyP signaling, which, ultimately, leads to the activation of proliferative markers in CRC. Overall, these findings might add iPolyP to the list of early biomarkers of CRC, together with DNA, RNA, and proteins, shown to prolong the overall survival of patients [35], thus potentially paving the way for the development of novel anticancer agents as a supplement to conventional chemotherapy.

## 2. Materials and Methods

### 2.1. Patients’ Samples

In this retrospective study, samples derived from 50 patients with CRC, who were candidates for surgical treatment, were studied. Patients provided written informed consent to the collection of blood samples for biomarker analysis under Prot. No. 397/C.E. of 16/09/2020 of the Local Ethics Committee “Gabriella Serio” IRCCS Istituto Tumori “Giovanni Paolo II,” Bari, Italy. Biopsy tissue specimens were provided by the Histopathology Unit of IRCCS “S. de Bellis.” The analyses were performed by comparing two different sections of a histological sample from the same patients, so as to identify a healthy portion (normal or peritumoral) and a diseased portion (pathological or tumoral). The inclusion criteria were patients with a confirmed diagnosis by colonoscopy, biopsy, or imaging studies, in whom surgery was considered beneficial. Patients with a grade of at least 2, i.e., cancer cells look more abnormal, were also considered. The patients to undergo surgery also had a good condition of health that was good enough to undergo surgery. This included having a reasonable performance status and no serious comorbidities that would significantly increase the risk of develop complications post-surgery. However, the patient had to be willing and able to consent to surgery after being informed of the risks, benefits, and potential outcomes. Samples collected in the operating room were temporarily stored in HypoThermosol FRS (for human cell and tissue preservation—Biolife solutions), sectioned, passed through liquid nitrogen, and stored dry in −80 °C within 3 h. 

### 2.2. Tissue Preparation

Patients (Pt) tissues, subdivided into two counterparts, peritumoral (PT) and tumoral (T), were tested for the amount of iPolyP, by fluorometric assay, and for the level of PCNA and TRPM8 by immunoblotting. Tissues were lysed using a tissue homogenizer (TissueLyser II, QIAGEN, Hilden, Germany; Cat. No. 85300) in Buffer III/Poly P Extraction Buffer, for iPolyP detection, described in detail in the inorganic polyphosphate detection assay section; or in the T-PER^TM^ Tissue Protein Extraction Reagent supplemented with Halt™ Protease and Phosphatase Inhibitor Single-Use Cocktail, EDTA-Free (100x for protein analysis, described in the Section 2.6.

### 2.3. Inorganic Polyphosphate Detection Assay

Inorganic polyphosphate was detected in tumoral and peritumoral tissues, using the Inorganic Polyphosphate Assay Kit (Fluorometric) (Abcam, Cambridge Biomedical Campus, Cambridge, UK; Cat. No. ab284528) following the relative manufacturers’ recommendations. Briefly, homogenized tissues were centrifuged at 10,000× *g* for 15 min at 4 °C and the clear supernatant was collected and treated with RNAse Positive Control/RNase, DNase, and Proteinase K. In total, 100 µL of the stock polyphosphate standard/iPolyP standard solution was mixed with 900 µL of water to prepare a 10 µM polyphosphate standard/iPolyP standard solution used to assay and plot the polyphosphate standard curve. To test 5 μL of each sample in triplicate, 96-well plates were prepared, adjusting the volume of each well to 50 μL using Polyphosphate Assay Buffer/iPolyP Assay Buffer. The reaction mix of 50 μL including 47 μL of Polyphosphate Assay Buffer and 3 µL of Polyphosphate Dye was added to each well, including standards and samples. The plates were incubated for 10 min at room temperature and the fluorescence of all wells was measured at Ex/Em = 415/550 nm at room temperature in endpoint mode. The calculation of the amount of inorganic polyphosphate was performed using the following equation: Amount of iPolyP (pmol/mg) = A × D/W × Vt/Va, where A = amount of iPolyP was calculated from the polyphosphate standard/iPolyP standard curve (pmol), D = sample dilution factor (D = 1, for undiluted samples), W = weight of the tissue used (in mg) or protein amount as determined by the protein assay (mg), Vt = total sample volume, and Va = volume of sample measured in the well. The protein amount (in mg) was determined using Bio-Rad (Bio-Rad Laboratories, Hercules, CA, USA; Cat. No. Hercules, CA, USA; Cat. No. 5000006EDU) protein assay dye reagent.

### 2.4. Cell Culture and Reagents

Human colorectal adenocarcinoma Caco-2 and SW620 cells were purchased from the American Tissue Culture Collection (ATCC, Manassas, VA, USA; Cat. No. HTB-37 and CCL-227, respectively). Human Colonic Epithelial Cells 1 transduced with CDK4 and Telomerase HCEC-1CT cells were purchased from Evercyte GmbH (Vienna, Austria; Cat. no. CkHT-039-0229). Caco-2 and SW620 cells were grown in Dulbecco’s modified Eagle’s medium (DMEM) (Thermo Fisher Scientific, Waltham, MA, USA; Cat. No. 11965092), supplemented with 10% fetal bovine serum (Thermo Fisher Scientific, Waltham, MA, USA; Cat. No. A5256701), 1 mM Sodium Pyruvate (Thermo Fisher Scientific, Waltham, MA, USA; Cat. No. 11360039), 25 mM HEPES (Thermo Fisher Scientific, Waltham, MA, USA; Cat. No. 15630056) and 100 U/mL antibiotic–antimycotic (Thermo Fisher Scientific, Waltham, MA, USA; Cat. No. 15240062). Human Colonic Epithelial Cells 1 transduced with CDK4 and Telomerase (HCEC-1CT) were grown in ColoUp medium ready to use (Evercyte GmbH, Vienna, Austria; Cat. No. MHT-039), supplemented with 100 U/mL antibiotic–antimycotic. All cell lines were maintained in a humidified atmosphere at 37 °C with 5% CO_2_. Cells were passaged and the medium was changed every other day.

### 2.5. Cellular Treatments

The HCEC-1CT, Caco-2, and SW-620 cell lines were seeded into 6-well plates (Corning, New York, NY, USA; Cat. No. 3516) at a density of 0.5 × 10^6^ cells/well in 2 mL of complete cell culture medium. Seeded cells were treated with 0.5 μM of sodium phosphate glass (iPolyP) (Sigma-Aldrich, St. Louis, MO, USA; Cat. No. S4379-500 mg); with 10 μM of the TRPM8 receptor inhibitor N-(3-Aminopropyl)-2-[(3-methylphenyl)methoxy]-N-(2-thienylmethyl)benzamide hydrochloride (AMTB) (Santa Cruz Biotechnology, Dallas, TX, USA; Cat. No. 926023-82-7); or in combination for 72 h. Dimethyl sulfoxide (DMSO) (Sigma-Aldrich, St. Louis, MO, USA; Cat. No. D8418-100 mL) was added to the control cells. The pharmacological inhibition of the iPolyP/TRPM8 axis was performed by adding AMTB to the HCEC-1CT, Caco-2, and SW620 cell lines. The pharmacological inhibition of the P2Y1 receptor was performed with Adenosine 2′,5′-diphosphate sodium salt (Santa Cruz Biotechnology, Dallas, TX, USA; Cat. No. SC-214495); RAGE receptor inhibition was performed with an antibody (Abcam, Cambridge Biomedical Campus, Cambridge, UK; Cat. no. ab235552). Cells were centrifuged at 1100 rpm for 5 min at 4 °C and washed with 1x sterile Dulbecco’s phosphate-buffered saline 1 (1x DPBS) (Thermo Fisher Scientific, Waltham, MA, USA; Cat. No. 14190-094) twice. Dried pellets were frozen at −80 °C or resuspended and lysed in 200 μL of T-PER^TM^ Tissue Protein Extraction Reagent (Thermo Fisher Scientific, Waltham, MA, USA; Cat. No. 78510) supplemented with Halt™ Protease and Phosphatase Inhibitor Single-Use Cocktail, EDTA-Free (100x) (Thermo Fisher Scientific, Waltham, MA, USA; Cat. No. 78443), for immunoblotting analysis.

### 2.6. Immunoblotting

Both tissue and cellular lysates were incubated on ice for 30 min and vortexed every 10 min. Samples were then centrifuged at 16,000 rpm at 4 °C for 20 min to clarify and precipitate insoluble debris. The total extracted proteins were assayed to measure concentrations using the Bio-Rad protein assay dye reagent concentrate (Bio-Rad Laboratories, Hercules, CA, USA; Cat. No. 5000006EDU). Then, proteins were mixed with 4 × Laemmli Sample Buffer (Bio-Rad Laboratories, Hercules, CA, USA; Cat. No. 1610747) and 10% of β-mercaptoethanol (Sigma-Aldrich, St. Louis, MO, USA; Cat. No. M6250-100 mL) and denatured at 95 °C for 5 min. In total, 25 μg of proteins was loaded onto precast polyacrylamide 4–20% gels (Bio-Rad Laboratories, Hercules, CA, USA; Cat. No. 4568094) and subsequently blotted on a polyvinylidene fluoride (PVDF) membrane (Bio-Rad Laboratories, Hercules, CA, USA; Cat. No. 1704156) using the trans-blot turbo transfer system (Bio-Rad Laboratories, Hercules, CA, USA; Cat. No. 1704150). Membranes were blocked using Pierce™ Protein-Free Blocking Buffer (Thermo Fisher Scientific, Waltham, MA, USA; Cat. No. 37571) for 1 h and stained overnight with primary antibodies. The next day, membranes were washed three times with 1X Tris-buffered saline (1x TBS) (Bio-Rad Laboratories, Hercules, CA, USA; Cat. No. 1,706,435, diluted in ddH_2_O to reach 1x)/TWEEN20 (Sigma-Aldrich, St. Louis, MO, USA; Cat. No. P9416-100 mL) incubated for 1 h with the respective horseradish peroxidase-conjugated secondary antibodies. Proteins were detected using the Clarity Max Western ECL Substrate (Bio-Rad Laboratories, Hercules, CA, USA; Cat. No. 1705062) and the signals were obtained using the Chemidoc MP Imaging System (Bio-Rad Laboratories, Hercules, CA, USA; Cat. No. 1708280). The following primary antibodies were used: anti-PCNA, 1:1000 (Santa Cruz Biotechnology, Dallas, TX, USA; Cat. No. SC-56); anti-TRPM8, 1:1000 (Thermo Fisher Scientific, Waltham, MA, USA; Cat. No. MA5-35474); anti-GAPDH, 1:1000 (Santa Cruz Biotechnology, Dallas, TX, USA; Cat. No. SC-47724); anti-β-Actin, 1:1000 (Cell Signaling Technology, Danvers, MA, USA; Cat. No. 4970S); anti-P2Y1-R, 1:1000 (Abcam, Cambridge Biomedical Campus, Cambridge, UK; Cat. No. ab168918); anti-RAGE-R, 1:1000 (Abcam, Cambridge Biomedical Campus, Cambridge, UK; Cat. No. ab216329). GAPDH and β-Actin were used as loading controls. The following secondary antibodies were employed: anti-rabbit IgG, HRP-linked antibody, 1:2000 (Cell Signaling Technology, Danvers, MA, USA; Cat. No. 7074S); goat anti-mouse IgG (H+L)-HRP conjugate (Bio-Rad Laboratories, Hercules, CA, USA; Cat. No. 1706516).

### 2.7. Knockdown of TRPM8 with siRNA

For the TRPM8-knockout experiments using siRNA, HCEC-1CT, Caco-2 and SW620 were electrophoresed with two TRPM8 Silencer siRNAs (Thermo Fisher Scientific, Cat. No. 4392420, ID: S35489 and ID: S35490, respectively) or with Silencer siRNA Negative Control (Thermo Fisher Scientific, Cat. No. 4390843). Electroporation was performed using the Neon™ NxT Electroporation System (Thermo Fisher Scientific, Cat. No. NEON1SK), according to the manufacturer’s recommendations.

### 2.8. Crystal Violet Assay

HCEC-1CT, Caco-2, and SW620 cells were seeded into 96-well plates (Corning, New York, NY, USA; Cat. No. 3599) at a density of 2 × 10^3^ cells/well in 100 μL of complete medium and left overnight to allow complete attachment. The next day (considered t = 0 h), the medium was removed and replaced with 100 μL of fresh serum-free medium to allow cell-cycle synchronization. Seeded cells in triplicate wells were treated and maintained in different conditions: 0.5 μM of iPolyP, or 10 μM of AMTB, or both in combination, for 96 h. Controls were cells treated with DMSO. After 96 h, cells were fixed by adding 100 μL/well of 4% paraformaldehyde (PFA) (Sigma-Aldrich, St. Louis, MO, USA; Cat. No. P6148-500G), pH 7.6, at room temperature for 30 min to the medium (obtaining a 2% PFA final concentration). Each well was subsequently stained with 0.02% of crystal violet solution (Sigma-Aldrich, St. Louis, MO, USA; Cat. No. HT90132-1 L) for 10 min and washed with cold water to eliminate surplus staining. Images were collected with a Nikon Ti2 inverted laser-scanning confocal microscope equipped with a 10x objective (0.25 numerical aperture), as described in the immunofluorescence and confocal microscopy section, in bright-field, and analyzed with NIS-Elements software (version 5.11.01) and ImageJ2 (version 2.14.0/1.54f) before complete solubilization. Finally, cell-bound crystal violet staining was redissolved in an aqueous solution containing 1% sodium dodecyl sulfate (SDS) (Sigma-Aldrich, St. Louis, MO, USA; Cat. No. L3771-100 G) at room temperature for 30 min and the absorbance was measured at λ = 595 nm with the iMark^TM^ Microplate Absorbance Reader (Bio-Rad Laboratories, Hercules, CA, USA; Cat. No. 168-1130).

### 2.9. Immunofluorescence Microscopy

In total, 2 × 10^5^/mL cells were grown in 35 mm petri dishes, No. 1.5 coverglass (MatTek, Ashland, MA, USA; Cat. No. P35G-1.5-14-C). Fixative, permeabilization, and blocking buffers were prepared in 1x DPBS. Cells were fixed with 4% PFA for 30 min at room temperature and then washed twice using 1X DPBS. Permeabilization was performed for 5 min at room temperature using 0.15% Triton X-100 (in 1x DPBS). Washing was performed to remove the permeabilization buffer. Cells were then blocked for 1 h at room temperature using blocking buffer (3% bovine serum albumin (BSA) (Sigma-Aldrich, St. Louis, MO, USA; Cat. No. A7030-100G)/1x DPBS). Cells were incubated with primary antibody overnight. Nuclei were stained using PureBlu DAPI (Bio-Rad Laboratories, Hercules, CA, USA; Cat. No. 1351303). Extensive washing steps were performed to remove unbound stain. Anti-Ki-67 (ready-to-use, Abcam, Cambridge Biomedical Campus, Cambridge, UK; Cat. No. ab16667) was used as the primary antibody. Images were collected with a Nikon Ti2 inverted laser-scanning confocal microscope equipped with Plan Apo 20x (0.75 numerical aperture) in bright-field and analyzed with NIS-Elements software (version 5.11.01) and ImageJ2 (version 2.14.0/1.54f). All fluorescence images were collected with a confocal Yokogawa spinning-disk on a Nikon Ti inverted microscope equipped with a Plan Fluor 20x (0.75 numerical aperture) lens; bright-field micrographs relative to CRC organoids were acquired with Plan Fluor 60x (0.85 numerical aperture); bright-field micrographs relative to spheroids were acquired with Plan Fluor 40x (0.60 numerical aperture) and 20x. Images were acquired with a Hamamatsu ORCA ER cooled CCD camera controlled with NIS-Elements software (version 5.11.01). Images were collected using an exposure time of 700 milliseconds. The gamma, brightness, and contrast were adjusted on displayed images (identically for comparative image sets) using NIS-Elements software (version 5.11.01). The Perfect Focus System was kept running for continuous maintenance of focus. DMEM without phenol red (Thermo Fisher Scientific, Waltham, MA, USA; Cat. No. 21063-045) was used during image acquisition.

### 2.10. CRC Tumor Organoids

CRC specimens from surgically resected tumor tissue were washed three times with 1x DPBS and digested with a collagenase/hyaluronidase mixture (Stemcell Technologies, Cat. No. 07912), diluted in HBSS solution (with CaCl_2_ and MgCl_2_, Thermo Fisher Scientific, Waltham, MA, USA; Cat. No. 14025-050) for 5 h under gentle rocking at 37 °C. A single-cell suspension was obtained and cells were embedded in a matrigel matrix basement membrane (Corning, New York, NY, USA; Cat. No. 356231) and cultured in IntestiCult-SF (ICT-SF) medium (Stemcell Technologies, Vancouver, BC, Canada; Cat. No. 100-0340) medium diluted 1:2 in Advanced DMEM/F-12 (Thermo Fisher Scientific, Waltham, MA, USA; Cat. No. 12634-010), supplemented with N-2 (Thermo Fisher Scientific, Waltham, MA, USA; Cat. No. 17502-048), B27 supplement (Thermo Fisher Scientific, Waltham, MA, USA; Cat. No. 12587-010), 0.01% bovine serum albumin, antibiotic–antimycotic, and HEPES. The medium was renewed every two days until the organoids were fully developed (7–10 days). Mature organoids were split using the TrypLE Select dissociation agent (Thermo Fisher Scientific, Waltham, MA, USA; Cat. No. A12177-01) according to [36] and the suspended cells obtained were re-cultured at a lower density in matrigel.

### 2.11. CRC Tumor Organoid Growth Test

Mature CRC tumor organoids were split to obtain single-cell suspensions. Cells were counted and 2000 cells were embedded in 40 µL of matrigel (with a matrigel density of 65%) in 96-well plates, cultured in ICT-SF medium, and allowed to grow for 10 days in the presence or absence of iPolyP at a concentration of 0.5 µM. The medium and iPolyP were replaced every 2 days. At the endpoint of the experiment, microscopic images of organoids were acquired in bright-field with the Nikon Confocal Microscope Eclipse Ti2. The growth rate of organoids was determined using the CellTiter 96 AQueous One Solution Cell Proliferation Assay (Promega Corporation, Fitchburg, WI, USA; Cat. No. G3580), according to the manufacturer’s recommendations.

### 2.12. Cell Lines-Derived Spheroids

In total, 1 × 10^3^ Caco-2, SW620, and HCEC-1CT cells were seeded into 3D low attachment 96-well cell culture plates (Corning, New York, NY, USA; Cat. No. 4520) to obtain 3D cell-line spheroids with 100 µL of growth medium in each well. Cells were kept in culture in the above-mentioned conditions. After seeding, all cell lines were treated with iPolyP and/or AMTB, while untreated cells were used as controls (untreated, UT). Spheroid cultures were observed at 24 and 96 h and maintained in a humidified incubator set to 37 °C and 5% CO_2_. After 96 h, UT and treated cells were fixed with a 4% paraformaldehyde solution in 1x PBS and used for microscopy analysis.

### 2.13. Gene Expression Analysis by Real-Time Quantitative Reverse Transcription PCR

Caco-2 cells were seeded into 6-well plates at a density of 0.5 × 10^6^ cells/well in 2 mL of complete cell culture medium. Seeded cells were treated with 0.5 μM of iPolyP, for 72 h; untreated cells were used as a control. Caco-2 cells were seeded into 6-well plates at a density of 0.5 × 10^6^ cells/well in 2 mL of complete cell culture medium. Seeded cells were treated with 0.5 μM of iPolyP, for 72 h; untreated cells were used as a control. Untreated cells were used as a control. RNA extraction was performed from frozen cell pellets using the RNeasy Mini Kit (QIAGEN, Hilden, Germany; Cat. No. 74104) according to the manufacturer’s recommendations. The RNA concentration was measured using a NanoDrop 2000c (Thermo Fisher Scientific, Waltham, MA, USA; Cat. No. ND-2000) and 2 μg total RNA was reverse transcribed to cDNA using a High-Capacity cDNA Reverse Transcription Kit (Thermo Fisher Scientific, Waltham, MA, USA; Cat. No. 4368814) following the relative protocol. Reactions were induced using the iTaq Universal SYBR Green Supermix (Bio-Rad Laboratories, Hercules, CA, USA; Cat. No. 1725124) and the validated human primers purchased from Bio-Rad (Hercules, CA, USA), with the following assay ID numbers: CCNA1, qHsaCID0008934; CCNB1, qHsaCID0010571; CCND1, qHsaCID0013833; GAPDH, qHsaCED0038674. Real-Time PCR analysis was performed with CFX96 Touch Deep Well Real-Time PCR Detection System (Bio-Rad Laboratories, Hercules, CA, USA; Cat. No. 3600037) and experiments were conducted for three times in triplicate. Relative expression was calculated using the 2^−ΔΔCt^ method.

### 2.14. Cell-Cycle Assay

Caco-2 and SW620 cells were seeded into 24-well plates (Corning, New York, NY, USA; Cat. No. 3524) at a density of 2 × 10^4^ cells/well in 100 μL of complete medium and left overnight to allow complete attachment. The following day, the medium was replaced with fresh serum-free medium to allow cell-cycle synchronization. Cells were treated with 0.5 μM of iPolyP, or with 10 μM AMTB hydrochloride, or both in combination for 72 h. Controls were cells treated with DMSO. After treatment, 150 μL of cell-clock dye, from the Biocolor Cell-Clock^TM^ Cell Cycle Assay (Ilex Life Science, Candler, NC, USA; Cat. No. C1000) kit, was added to the center of each well and incubated for 1 h at 37 °C, according to the manufacturer’s recommendations. The culture medium added with the reagent was softly discarded and replaced with 200 μL of fresh medium. Micrographs of live cells were acquired using a Nikon Confocal Microscope Eclipse Ti2 in bright-field equipped with a Plan Fluor 20X (0.75 numerical aperture) lens, and all experiments were conducted in triplicate. The calculation of phase percentages was obtained by analysis made with ImageJ2 software. The redox dye used in this test is taken up by live cells and the outcome is a color-defined cell-cycle stage; each color was associated with cells in G0–G1 (yellow staining), S (green staining), or G2 and M (dark green/blue phases).

### 2.15. Statistical Analysis

Patients’ characteristics are reported as the mean and standard deviation (M ± SD), and as frequencies and percentages (%) for categorical variables. To compare iPolyP values between groups, the Mann–Whitney rank test was used for continuous variables. The Spearman rank correlation coefficient was used to test the strength and direction of associations between iPolyP and TRPM8/PCNA. When testing the null hypothesis of no association, the two-tailed probability level of error was set at 0.05. All statistical computations were made using StataCorp. (2021) Stata Statistical Software: Release 18 (StataCorp LLC, College Station, TX, USA), while RStudio (version “Chocolate Cosmos” Release, Posit PBC, Boston, MA, USA) was used for the plots.

## 3. Results

### 3.1. Human Colorectal Cancer Tissue Displays Enhanced Levels of iPolyP That Are Correlated with the Proliferation Marker PCNA

To estimate the iPolyP levels in patients with colorectal cancer (CRC), we recruited 50 patients undergoing surgery. Males (m) had a higher prevalence of CRC (52.00%) than females (f) (48.00%) at an average mean age of 71.53 ± 11.19 years. The tumor–nodes–metastasis (TNM) and grading classification of each subject is shown in Table 1. From each subject, we collected the peritumoral portion and tumoral counterpart to perform iPolyP level detection. Intriguingly, the concentration of iPolyP was higher in tumoral tissue than in the peritumoral section, with a statistically significant difference (373,097.70 ± 210,216.20 pmol/mg vs. 166,102.70 ± 124,618.80 pmol/mg, respectively), suggesting a putative role of iPolyP in promoting the tumorigenicity of CRC (Figure 1). Moreover, in the same analyzed samples, tumoral tissue exhibited elevated levels of the proliferating cell nuclear antigen (PCNA) protein, a well-known marker of DNA replication and cellular proliferation, when compared to the matched peritumoral tissue (Figure 2A). In addition, a strong positive correlation between iPolyP and PCNA expression was found (ρ = 0.44, **** *p* < 0.0001) (Figure 2B), which further suggests a synergistic pro-tumorigenic contribution of iPolyP to CRC development.

### 3.2. The iPolyP–TRPM8 Signaling Axis Sustains Colorectal Cancer Cell Proliferation

To determine whether iPolyP has pro-neoplastic properties and, if so, which receptor-binding partner signals to the colorectal cancer milieu, we initially screened biopsies from the same cohort of enrolled subjects, for the P2Y1 receptor (P2Y1-R), RAGE receptor (RAGE-R), and TRPM8 channel receptor, all well-recognized iPolyP receptors. Among these, only TRPM8 appeared to be overexpressed in the tumoral fraction compared with the peritumoral one (Figure 3A) in all samples, while the P2Y1-R and RAGE-R levels were not different (Appendix A). Moreover, similarly to PCNA, we found a strong positive correlation between the concentration of iPolyP within the tumoral tissues and TRPM8 receptor expression (ρ = 0.48, **** *p* < 0.0001) (Figure 3B), supporting the hypothesis that iPolyP may primarily transduce through the TRPM8 receptor in the CRC context. Following these findings, we firstly explored whether the presence of iPolyP could somehow encourage the proliferation of colorectal cancer cells by interacting with the TRPM8 receptor. With an in vitro approach, we demonstrated that the administration of iPolyP induces the expression of PCNA in Caco-2 (Figure 4A) and in SW620 (Appendix A), non-metastatic and metastatic colorectal cancer cell lines, respectively, without altering the TRPM8 levels (Figure 4A). Surprisingly, no detectable effect of iPolyP was observed at the PCNA level in HCEC-1CT cells, the cytogenetically normal and nontumorigenic colon-derived cell line (Appendix A). Following pharmacological inhibition or genetic abrogation of the TRPM8 receptor, we could revert PCNA expression to levels comparable to those observed in the untreated samples, in both the colorectal cancer cell lines examined (Figure 4A and Appendix A), while no difference was observed in HCEC-1CT cells upon iPolyP challenge (Appendix A). Direct evidence of iPolyP–TRPM8 axis-mediated cellular proliferation was obtained with crystal violet staining, a quantitative assay that discriminates live versus dead cells based on the DNA intercalating dye, proportional to the number of adherent cells. After 96 h, iPolyP revealed a marked propensity to promote Caco-2 and SW620 proliferation. Furthermore, as expected, by antagonizing or knocking down the TRPM8 receptor, we flattened the proliferation rate (Figure 4B,C and Appendix A). This significant highly proliferative phenotype is not discernible in HCEC-1CT cells upon iPolyP treatment, whose division rate remains unaltered (Appendix A). iPolyP does not seem to influence the expression levels of the P2Y1 and RAGE receptors in HCEC-1CT, Caco-2, or SW620 cells (Appendix A), thus excluding a positive feedback mechanism mediated by iPolyP underpinning the molecular regulation of its receptors. Proliferation assays performed on Caco-2 and SW620 with the P2Y1 or RAGE inhibitor demonstrated that iPolyP mainly triggers colorectal cancer cell expansion by binding to TRPM8, as the hindrance of these pathways did not affect the proliferation induced by iPolyP (Appendix A). To corroborate the proliferative propensity of iPolyP, we performed fluorescence microscopy experiments targeting Antigen Kiel 67, known as Ki-67, a second common marker for proliferation, in our model cell lines, whose outcome revealed PCNA alterations. In particular, upon iPolyP administration, Caco-2 and SW620 cells, but not HCEC-1CT, displayed an enhanced level of Ki-67, which was suppressed following TRPM8 inhibition or when silenced (Appendix A). Moreover, SW620 cells showed self-organization into spheroids, whose morphology appeared enlarged in the presence of iPolyP. All together, these data deliver an important take-home message regarding the capability of iPolyP to promote colorectal cancer cell growth by engaging the TRPM8 receptor.

### 3.3. iPolyP Encourages CRC Patient-Derived Organoids’ Growth and Caco-2- and SW620 Cells-Derived Spheroids’ Formation

Based on the above findings, we recapitulated the proliferative task of iPolyP with tests of CRC patient-derived organoids’ growth. We generated organoids from two CRC subjects and incubated them with iPolyP. As shown in Figure 5A,B, the growth rate of the organoids in the presence of iPolyP was significantly higher than in those left untreated, for both independent experiments (Figure 5A,B). Moreover, these data reflected the evidence observed with Caco-2- and SW620 cells-derived spheroids; in particular, after 96 h of incubation with iPolyP, the diameter of the spheroids was double that of the untreated ones. The presence of the TRPM8 inhibitor abrogated the proliferative effect due to iPolyP. Finally, we could appreciate no change in HCEC-1CT cells (Figure 5C,D), confirming our conclusions regarding the role of iPolyP in CRC.

### 3.4. iPolyP Induces ccnb1 Expression in the Caco-2 Cell Line and Drives the Cells into M Phase via the TRPM8 Receptor

To further explore the molecular mechanism by which inorganic polyphosphate drives the expansion of colorectal cancer cells, we performed real-time quantitative reverse transcription PCR experiments on Caco-2 cells-derived RNA, 72 h post iPolyP treatment, probing genes involved in different phases of the cells cycle. Although we noticed a considerable increase, in terms of the number of copies of transcripts related to genes like *ccnd1* (cyclin D1), *ccna1* (cyclin A1), *ccne1* (cyclin E1), and cyclin-dependent kinases-2, -4, and -6 (*cdk2*, *cdk4,* and *cdk6*, respectively), mainly involved in the G1 and S/G2 phases, a significant signal amplification was observed for the *ccnb1* transcript, which codes for the mitotic protein cyclin B, a master regulator of the G2/M phase (Figure 6A and Appendix A). Figure 6B summarizes the most relevant cyclins linked to different phases of the cell cycle. To make a cellular interpretation of the gene expression data, we performed a “cell-clock” cell-cycle assay on Caco-2 and SW620 cells exposed to iPolyP for 72 h. This assay consists of live-cell detection employed to monitor the four major phases of the mammalian cell cycle through a redox dye. Following dye uptake, a distinct color change occurs within cells, denoting the specific G1, S, G2, and M phases of the cycle. In detail, it turns yellow in G1, green in S/G2, and blue in M phase. A significantly high percentage of blue-colored cells was spotted following iPolyP administration compared to the control sample, the latter being comparable to the TRPM8 inhibitor- or iPolyP + TRPM8 inhibitor-treated cells (Figure 6C,D). Overall, these data strengthen our interpretation of a pro-tumorigenic role of iPolyP in colorectal cancer cells.

## 4. Discussion

Despite the enormous efforts made striving towards the development of new therapeutical strategies targeting the molecular complexity discernible in colorectal cancer [37], CRC remains one of the deadliest human neoplasms due to its complex interplay of factors including advanced age, environmental influences, and genetic alterations [38,39]. These elements underlie each stage of tumorigenesis, thus limiting the effectiveness of specific therapy [38]. Cancer cell metabolism thrives best in a hypoxic environment and relies on alternative energy sources to survive and proliferate [40,41]. The linear polymer composed of several up to hundreds of phosphate residues, named inorganic polyphosphate, fits the role of energy supplier well, with its phosphoanhydride ATP-like bonds [42]. Not surprisingly, in fact, the involvement of iPolyP in cancer cell proliferation and tumor progression has been recently proposed and demonstrated in vitro [24]. In mammals, enzymatically synthesized iPolyP has two known sources, bacterial and human. Bacterial-derived iPolyP ranges from 100 to 1000 units thanks to the high processivity of the enzyme polyphosphate kinase 1 (*ppk1*), which uses ATP as a substrate [43,44,45]. Human-derived iPolyP shows an average of ~60–100 phosphate residues, and no known kinase has thus far been linked [18,46,47]. Understanding the biological source of iPolyP in the context of CRC is a fundamental question, as it could help to design a targeted therapeutic strategy. The different number of orthophosphate residues might affect, in fact, the proliferative, pro-inflammatory, and pro-tumorigenic tasks of iPolyP. Moreover, it is largely unknown whether iPolyP derives from peculiar strains within an altered intestinal flora (a condition termed dysbiosis) which contribute most strongly to iPolyP synthesis. However, emerging studies, assisted by state-of-the art methodologies, are revealing the influence of a dysbiotic phenotype in CRC onset and progression, which is bringing to light an exponential number of novel biomarkers [48,49]. In this context, we studied whether iPolyP, of either bacterial or human origin, could somehow foster the pathogenesis of CRC. By applying in vitro and ex vivo approaches, we demonstrated for the first time that iPolyP facilitates CRC cell proliferation by markedly engaging the TRPM8 receptor channel. This conclusion is based on the following evidence, summarized in Figure 7: (i) tumoral tissues isolated from CRC subjects betrayed elevated levels of iPolyP, proportional to the amount of the PCNA proliferation marker; (ii) iPolyP governs CRC cell lines’ proliferation by interacting with the TRPM8 receptor, found to be positively correlated with the concentrations of iPolyP within tumoral tissues; (iii) ex vivo experiments, performed on CRC patient-derived organoids and in vitro 3D cell culture, showed a significantly higher growth rate in the presence of iPolyP compared to untreated samples or samples in which TRPM8 was antagonized; (iv) the iPolyP–TRPM8 axis engagement triggers the expression of *ccnb1*, whose corresponding protein product is Cyclin B, which drives the cells into the mitotic phase of the cell cycle. Understanding the source of iPolyP is becoming decisive to target the proliferative pathway. Several studies in the field of microbiology are nowadays focusing on pathogenic bacteria, such as *Salmonella enterica*, with a particular emphasis on the mechanism employed by iPolyP in the onset of disparate mammalian diseases [50,51,52]. Intriguingly, a recent article led by Boyineni and colleagues showed that cancer cells are capable of endogenously producing iPolyP which fosters the metabolism under their period of starvation, as a result of an insufficient tumor vascular supply [53]. Hence, although our findings point out, for the first time, the presence of iPolyP within tumoral samples of CRC-affected individuals, a significant limitation in the study is the lack of the information regarding the origin, as it can be either from active platelets infiltrating the tumor, cancer cells, or dysbiotic intestinal microbiota. This knowledge will allow the development of a targeted and specific therapy. Thus, ongoing mass spectrometry experiments in our laboratory aim firstly to elucidate the provenience of iPolyP within the CRC context, steering research toward the appropriate in vivo models targeting either the bacterial or human kinase responsible for inorganic polyphosphate biosynthesis. In addition, we will apply these findings to other cancer types and determine the role of iPolyP in different experimental settings. This might potentially lead to a classification of iPolyP-sensitive neoplasms or limit its activity solely to CRC. Finally, we will consider the suitability of iPolyP as a novel, non-invasive biomarker for the early detection of CRC. Although the molecular pathway needs to be fully interpreted, and more experimental evidence is required to confirm our hypothesis, these results uncover a novel, functional axis in the CRC background, shedding light on new directions for study and paving the way for the development of new therapeutic strategies for CRC patients.

## 5. Conclusions

The findings regarding the iPolyP–TRPM8 axis suggest potential novel therapeutic strategies, such as targeted therapies that minimize TRPM8 activity, in combination with the conventional chemotherapy protocols applied nowadays. In vivo experimental CRC models, involving TRPM8-deficient mice or mice treated with a TRPM8 inhibitor, to support our conclusions are currently under study. In addition to potential therapeutic applications, our research also aims to explore whether iPolyP detection may represent a novel approach for the screening of CRC. An early diagnosis of CRC requires a rather invasive method, generally involving biopsy samples taken at certain time intervals. However, this method poses several challenges and limitations, such as invasiveness; limited accessibility (for tumors located nearby inaccessible areas); and resource intensiveness, since it requires specialized skills; as well as producing a temporal snapshot, which provides a picture of the tumor at a single point in time and might not represent the full heterogeneity of the tumor or catch the dynamic changes it undergoes during treatment. Liquid biopsy offers a valid alternative, without employing invasive procedures, to monitor the disease status, allowing disease detection and monitoring of the progression or response to therapy. Bona fide non-invasive biomarkers should possess high sensitivity properties and be quantifiable within fluids, including serum, sweat, urine, saliva, or fecal samples. Moreover, they must reflect the body’s responses to tumor progression and/or therapy. Our aim is therefore to test the efficacy of iPolyP as a novel diagnostic and prognostic biomarker to be detected through liquid biopsy or by screening stool samples of CRC-affected subjects. Thus, if successful, this research could pave the way for the development of novel, non-invasive detection methods that can offer long-term benefits and improve healthcare and patients’ lives with a significant social impact.

## Figures and Tables

**Figure 1 cancers-16-03326-f001:**
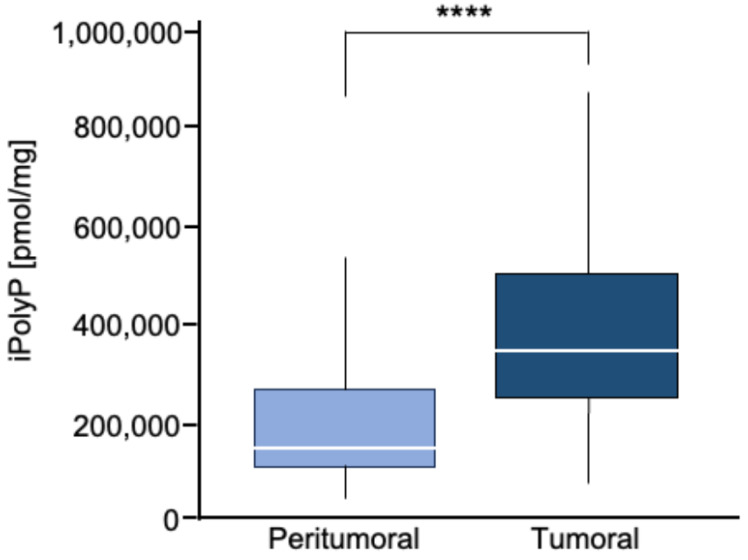
Box plot of iPolyP distribution, stratified by peritumoral or tumoral patients. A statistical difference was found (median value of 373,097.70 ± 210,216.20 pmol/mg for tumoral tissue vs. 166,102.70 ± 124,618.80 pmol/mg for the peritumoral counterpart, **** *p* < 0.0001). Analysis was performed by the Mann–Whitney rank test.

**Figure 2 cancers-16-03326-f002:**
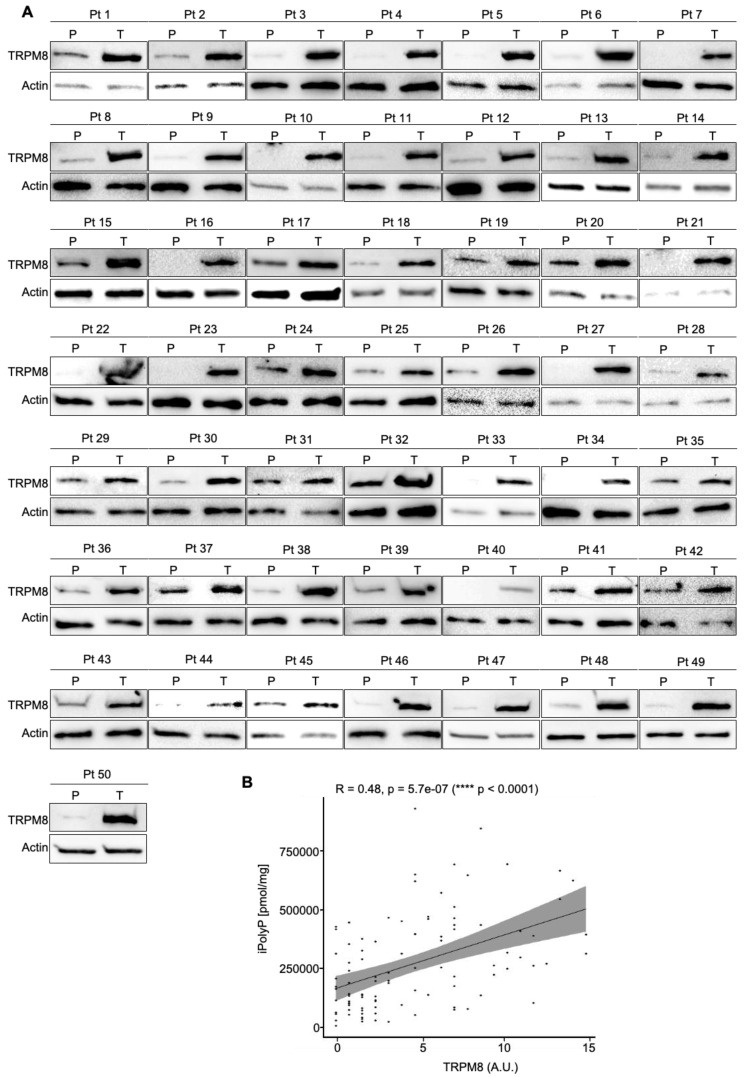
CRC displays a high level of PCNA. (**A**). Cellular extracts from 50 human biopsies, where the tumoral sample (T) was plotted against the peritumoral (P) counterpart of the same patient (Pt), were analyzed by immunoblotting for the PCNA expression level. Actin was used as a loading control for the normalization. The uncropped bolts are shown in Appendix A. (**B**). The Spearman correlation between iPolyP and PCNA in the total cohort, denoting a strong positive correlation (**** *p* < 0.0001). Fold changes versus peritumoral (P), normalized to 1.

**Figure 3 cancers-16-03326-f003:**
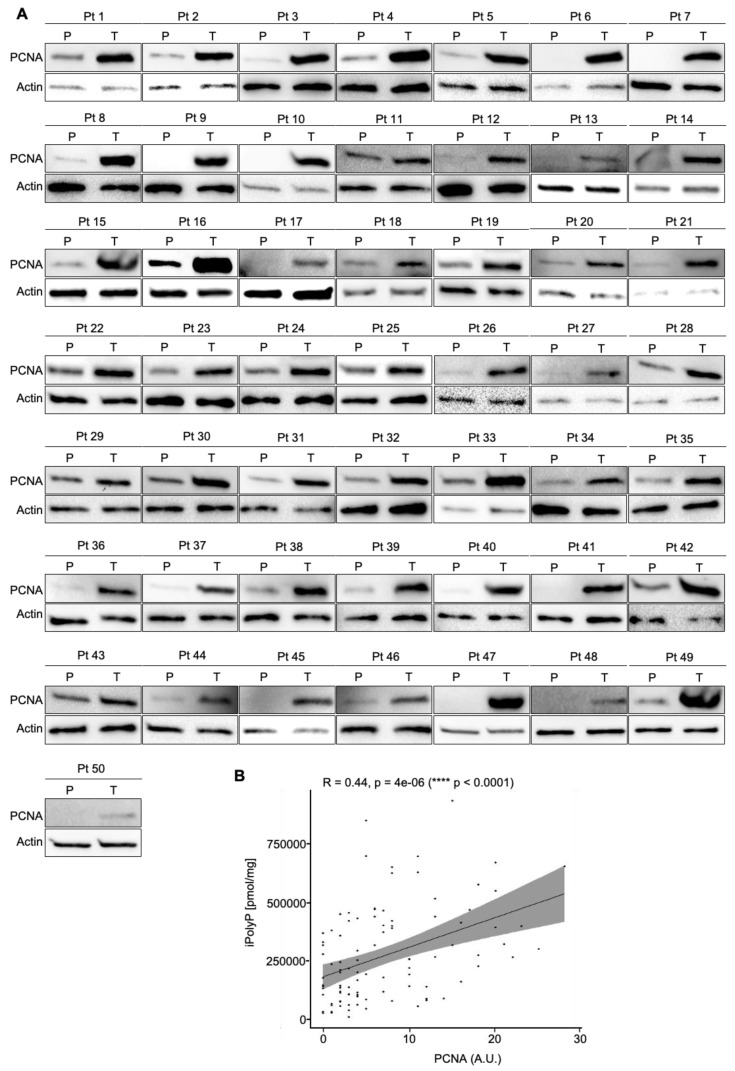
CRC displays a high level of TRPM8 receptor. (**A**). Cellular extracts from 50 human biopsies, where the tumoral sample (T) was plotted against the peritumoral (P) counterpart of the same patient (Pt), were analyzed by immunoblotting for the TRPM8 expression level. Actin was used as a loading control for the normalization. The uncropped bolts are shown in Appendix A. (**B**). The Spearman correlation between iPolyP and TRPM8 in the total cohort, denoting a strong positive correlation (**** *p* < 0.0001). Fold changes versus peritumoral (P), normalized to 1.

**Figure 4 cancers-16-03326-f004:**
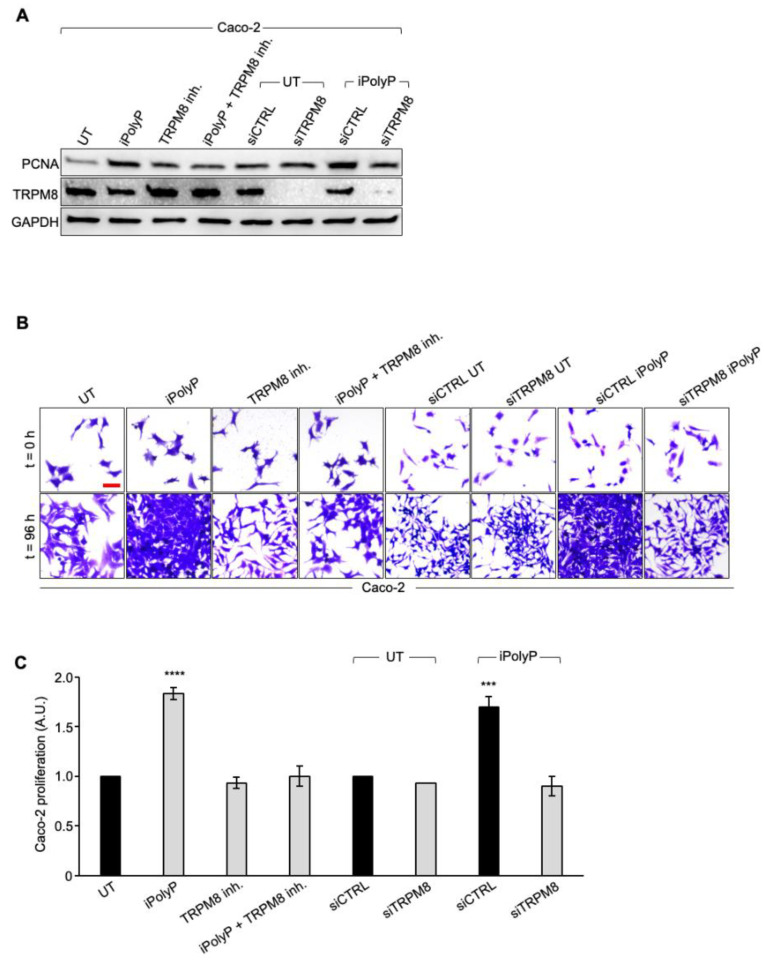
iPolyP enhances PCNA expression and promotes Caco-2 colorectal cancer proliferation. (**A**). Cellular extracts from WT and siRNA-mediated TRPM8 knockdown Caco-2 cell lines were analyzed by immunoblotting for the PCNA expression level. GAPDH was used as a loading control. The uncropped bolts are shown in Appendix A. (**B**). Representative micrographs of the crystal violet assay performed on WT and siRNA-mediated TRPM8 knockdown Caco-2 cell lines upon treatment for 96 h with iPolyP, a TRPM8 inhibitor, or both. Scale bar, 10 µm. Images are representative of three independent experiments. (**C**). Statistical analysis of the crystal violet assay by Student’s *t*-test, respectively, for panel (**C**) (*** *p* < 0.001 and **** *p* < 0.0001). Fold changes versus control, untreated (UT), normalized to 1. Data are presented as the mean ± SD for triplicate wells from three independent experiments.

**Figure 5 cancers-16-03326-f005:**
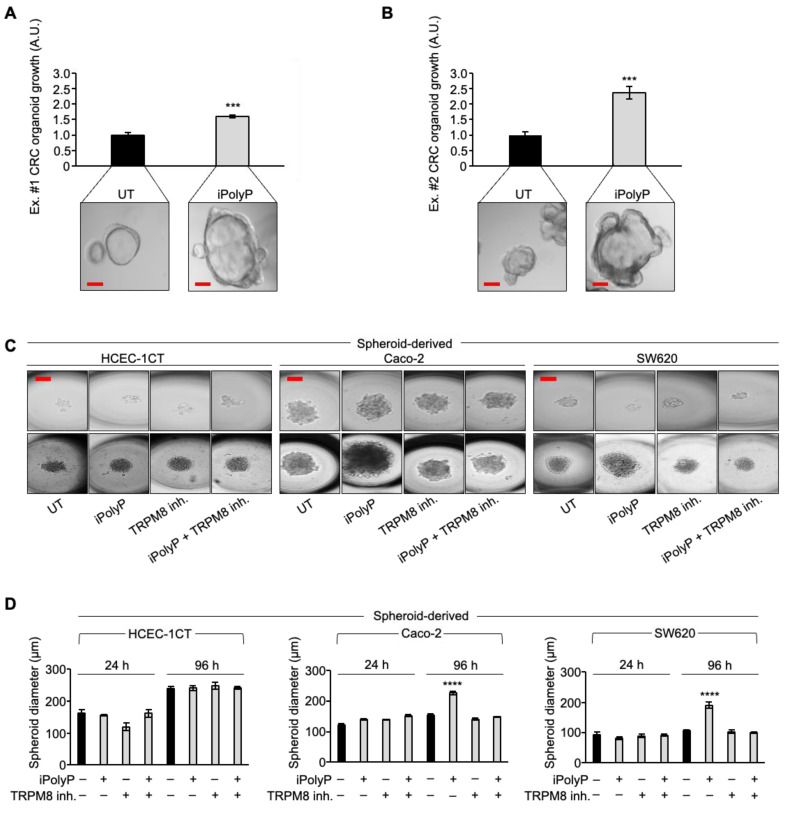
iPolyP promotes colorectal cancer patient-derived organoids and Caco-2- and SW620-derived 3D spheroids. (**A**,**B**). Two independent experiments of a CRC patient’s derived organoids, assessed by light microscopy, in the absence of iPolyP (UT) or incubated for 10 days in the presence of iPolyP. Scale bar, 100 µm. Fold changes versus control, untreated (UT), normalized to 1. Statistical analysis was performed by Student’s *t*-test (*** *p* < 0.001). (**C**). Representative bright-field images of 24 h- and 96 h-induced spheroid formation derived from the HCEC-1CT, Caco-2, and SW620 cell line, respectively, upon treatment with iPolyP, a TRPM8 inhibitor, or both for 96 h. Scale bar, 100 µm. Images are representative of three independent experiments. (**D**). Quantification relative to panel (**C**). Fold changes versus control, untreated (UT). Statistical analysis was performed by Student’s *t*-test (**** *p* < 0.0001). Data are presented as the mean ± SD for triplicate wells from three independent experiments. The uncropped bolts are shown in Appendix A.

**Figure 6 cancers-16-03326-f006:**
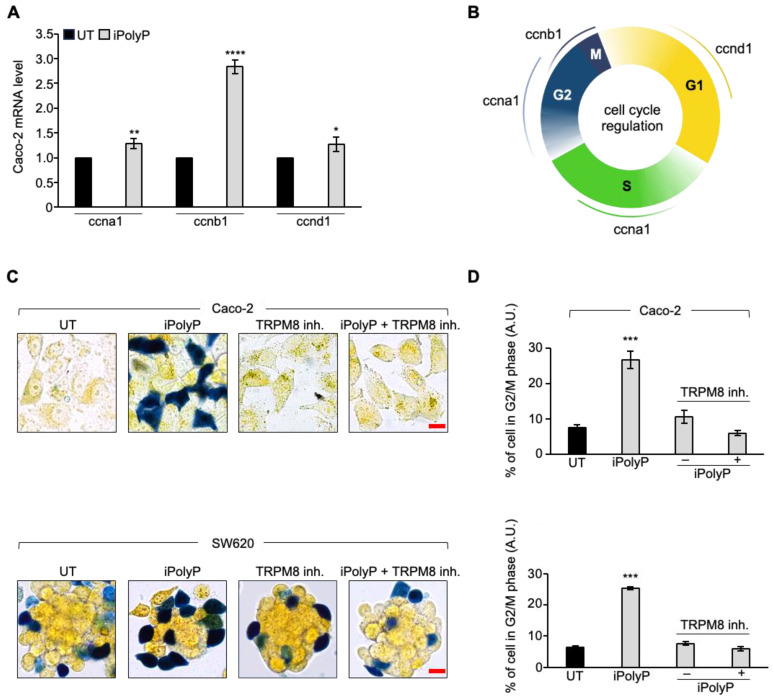
iPolyP drives cells into G2/M phase. (**A**) Real-time PCR on the iPolyP-treated Caco-2 cell line for 72 h on cyclins implicated in different phases of the cell cycle; untreated (UT) samples were normalized to 1. * *p* < 0.05; ** *p* < 0.01; **** *p* < 0.0001. (**B**) Sketch representing cyclins-dependent cell-cycle regulation consisting of Gap 1 (G1), synthesis (S), Gap 2 (G2), and mitosis (M). Figure was created with BioRender. (**C**) Representative micrographs of the cell-cycle assay on Caco-2 (**upper panel**) and SW620 (**lower panel**) cell line treated for 72 h with iPolyP, a TRPM8 inhibitor, or both. Scale bar = 10 µm. Images are representative of three independent experiments. (**D**) Percentage of cells in G2/M phase. Statistical analysis was performed by Student’s *t*-test (*** *p* < 0.001). Fold changes versus control, untreated (UT). Data are presented as the mean ± SD for triplicate wells from three independent experiments.

**Figure 7 cancers-16-03326-f007:**
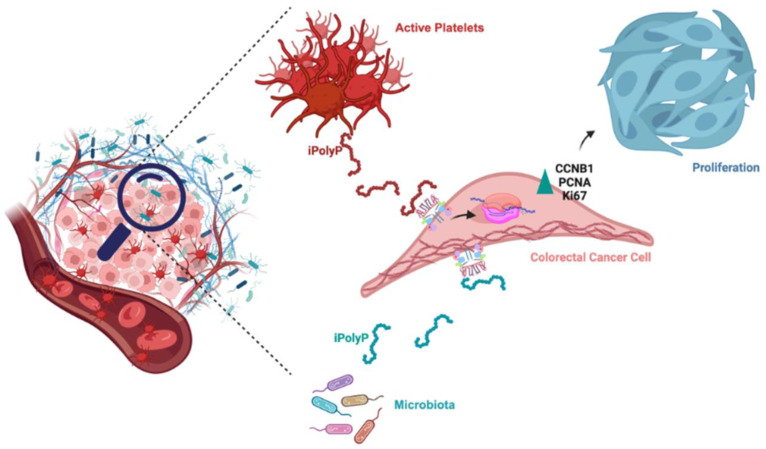
Schematic of the role of iPolyP derived from platelets and microbiota on macrophages and CRC cancer cells. The iPolyP–TRPM8 axis stimulates colorectal cancer expansion by enhancing the level of the *ccnb1* gene, which coordinates the M phase of the cell cycle, alongside the expression of two bona fide proliferative markers, PCNA and Ki-67.

**Table 1 cancers-16-03326-t001:** Patients index with age at the time of the surgery, sex, stage, tumor–node–metastasis (TNM), and grading.

Patient	Age	Sex	Diagnosis	TNM	Grading
1	58	f	CRC	T1N0MX	G3
2	58	m	CRC	T1N0MX	G2
3	58	f	CRC	T1N0MX	G2
4	81	f	CRC	T2N0MX	G3
5	72	m	CRC	T2N0MX	G2
6	78	f	CRC	T2N0MX	G3
7	76	m	CRC	T2N0MX	G3
8	82	m	CRC	T2N0MX	G2
9	85	m	CRC	T2N0MX	G2
10	73	f	CRC	T2N0MX	G3
11	82	f	CRC	T2N0MX	G2
12	81	m	CRC	T2N1aMX	G2
13	70	m	CRC	T2N1aMX	G3
14	60	f	CRC	T2N1aM1	G3
15	71	m	CRC	T3N0MX	G2
16	75	m	CRC	T3N0MX	G2
17	59	f	CRC	T3N0MX	G3
18	69	m	CRC	T3N0MX	G3
19	89	m	CRC	T3N0MX	G3
20	72	f	CRC	T3N0MX	G3
21	59	f	CRC	T3N0MX	G3
22	77	f	CRC	T3N0MX	G3
23	70	m	CRC	T3N0MX	G2
24	65	m	CRC	T3N0MX	G3
25	43	f	CRC	T3N0MX	G2
26	76	m	CRC	T3N0MX	G2
27	65	f	CRC	T3N0MX	G2
28	74	m	CRC	T3N0MX	G2
29	69	f	CRC	T3N0MX	G3
30	71	f	CRC	T3N0MX	G2
31	70	f	CRC	T3N0M1b	G2
32	88	m	CRC	T3N1aMX	G2
33	74	m	CRC	T3N1aMX	G2
34	80	f	CRC	T3N1aMX	G3
35	74	m	CRC	T3N1aMX	G3
36	70	f	CRC	T3N1aMX	G3
37	67	f	CRC	T3N1aMX	G3
38	78	m	CRC	T3N1bMX	G2/G3
39	79	m	CRC	T3N1bMX	G3
40	77	m	CRC	T3N1bMX	G3
41	83	m	CRC	T3N2aMX	G3
42	76	m	CRC	T4aN0MX	G3
43	56	m	CRC	T4aN1MX	G3
44	35	m	CRC	T4aN1bMX	G3
45	86	m	CRC	T4aN2bMX	G2
46	93	f	CRC	T4aN2bM1c	G3
47	81	f	CRC	T4bN0MX	G2
48	71	f	CRC	T4bN0MX	G2
49	53	f	CRC	T4bN0MX	G2
50	74	f	CRC	T4bN0MX	G2

## Data Availability

The original raw data presented in the study are openly available at Appendix A.

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
