# Peer review of "Inorganic Polyphosphate Promotes Colorectal Cancer Growth via TRPM8 Receptor Signaling Pathway"

_cancers, 2024, doi:10.3390/cancers16193326_

Round 1

Reviewer 1 Report

Comments and Suggestions for Authors

13 Sep 2024

Dear Editor-in-Chief

Thank you for your Email regarding the review of Manuscript ID: cancers-3223515, entitled: "Inorganic polyphosphate promotes colorectal cancer growth via 2 TRPM8 receptor signaling pathway”. The manuscript could be acceptable for publication after major revisions as follows;

Comments:

Abstract

1. The term "underestimated" may not be the most appropriate here. Consider using "underexplored" or "less studied" to better convey the idea that iPolyP has not been thoroughly investigated in this context.

 2.  While you mention that iPolyP is "ubiquitous in all forms of life," it may be beneficial to briefly state its biological significance or functions to provide context for readers unfamiliar with the molecule.

 3.  When mentioning TRPM8, consider briefly explaining its general function or relevance in CRC to enhance understanding.

 4. Specify what these approaches entail (e.g., cell lines used, types of ex-vivo tissues). This will help readers understand the experimental design better.

 5.  Ensure consistent formatting for terms like "in vitro" (use italics) and "ex vivo," as this enhances professionalism and readability.

Introduction

6.  Some sentences are long and complex, which can hinder readability. For example, "Individuals bearing CRC display backgrounds that cover a wide range of genetic or epigenetic alterations" could be simplified to "Patients with CRC exhibit a wide range of genetic and epigenetic alterations."

 7. Improve transitions between sentences and paragraphs to enhance flow. For instance, after discussing genetic mutations, a clearer transition to lifestyle factors would help maintain coherence.

 8. Use of "belong": The phrase "TRPM8 belongs to the transient receptor potential (TRP) cation channel superfamily" should be corrected to "TRPM8 belongs to the transient receptor potential (TRP) cation channel superfamily."

 9. "iPolyP architecture makes it an obvious participant": The term "obvious" is subjective. Consider using "suggests that iPolyP plays a significant role in energy metabolism."

10. When introducing acronyms like RAGE and TRPM8, consider providing a brief description or function to help readers unfamiliar with these terms.

 11. Mechanisms: The introduction mentions that "mechanistically, a number of queries remain unaddressed." It would be beneficial to specify some of these queries to emphasize the significance of your research.

 12. Relevance of Citations: Some references (e.g., [15-20]) are broad. Consider specifying key studies or findings related to iPolyP's role in inflammation or tumorigenesis for better context.

 13. While the introduction identifies a gap in knowledge regarding iPolyP's role in cancer, it could more explicitly state the specific aims or hypotheses of your study at the end of the introduction. This would provide a clearer roadmap for readers.

 14. "can spontaneously arise from non-living matter": This statement could be misleading without context. Specify how this occurs or provide a citation to support this claim.

 15. The final sentences should succinctly summarize the study's objectives. For example, "In this study, we aimed to quantify iPolyP levels in CRC tissues and investigate the role of TRPM8 as a mediator of iPolyP signaling."

Method:

16. While you mention that patients were candidates for surgical treatment, it would be beneficial to include specific inclusion and exclusion criteria to provide context on how patients were selected for the study.

 17. While you mention studying samples from 50 patients, it would be helpful to provide justification for this sample size. Was it based on power analysis or previous studies?

 18. Include information on how samples were stored (e.g., temperature, duration) before analysis to ensure reproducibility.

 19. Ensure that terms like "Peritumoral" and "Tumoral" are consistently capitalized throughout the section.

20. It may be helpful to briefly justify the chosen concentrations of iPolyP and AMTB based on previous studies or preliminary experiments.

Result:

21. All figures need to be revised to clearly show scale bars.  In addition, the quality of microscopic images needs to be improved. 

Discussion

22. Some sentences are lengthy and complex, making them difficult to follow. For example, the sentence starting with "The CRC elusiveness is primarily due to..." could be broken down for clarity.

 23. Simplify sentences and use shorter phrases to enhance readability. For instance:

     - "CRC remains one of the deadliest human neoplasms due to its complex interplay of factors including advanced age, environmental influences, and genetic alterations."

   24.  Maintain consistency in terminology and ensure all abbreviations are defined clearly.

   25.  The section mentions "evidence is still minor" regarding iPolyP's role in cancer, which may undermine the significance of your findings.

   - Suggestion: Instead of stating the evidence is limited, emphasize the novelty of your findings and how they contribute to the existing body of knowledge.

   26. The transition from discussing iPolyP to its sources (bacterial vs. human) feels abrupt. More context on why this differentiation is important could improve flow.

   - Suggestion: Introduce a transitional sentence that links the importance of understanding the source of iPolyP to its potential therapeutic implications in CRC.

   27. Lack of Future Implications: While the discussion touches on potential therapeutic strategies, it lacks a clear direction for future research.

   - Suggestion: Conclude with a paragraph outlining potential next steps in research, such as exploring specific pathways influenced by iPolyP or investigating its role in other cancer types.

Conclusion

28. The phrase "involvement of the iPolyP-TRPM8 axis may offer novel therapeutic options" could be more precise about what these options entail.

 - Suggestion: Clarify what specific therapeutic options are being considered. For example, "The findings regarding the iPolyP-TRPM8 axis suggest potential novel therapeutic strategies, such as targeted therapies that enhance TRPM8 activity."

29. The statement about in-vivo experimental models being under study lacks detail. It would strengthen the discussion to mention preliminary results or specific methodologies.

 - Suggestion: Include a brief overview of the experimental design or any preliminary findings to bolster the claim.

30. The sentence describing early diagnosis methods could benefit from more detail regarding their limitations.

 - Suggestion: Expand on the invasive nature of current biopsy methods and why they pose challenges for early diagnosis, such as patient discomfort or delayed results.

31. The aim to test iPolyP as a diagnostic and prognostic biomarker could be more specific about what aspects will be evaluated.

 - Suggestion: Specify what parameters will be assessed in the study, such as sensitivity, specificity, or correlation with disease stage.

32. The transition from discussing therapeutic options to diagnostic methods feels abrupt.

 - Suggestion: Use transitional phrases to connect these ideas more smoothly. For example, "In addition to potential therapeutic applications, our research also aims to explore..."

33. - Lack of Forward-Looking Statements: The discussion could benefit from mentioning potential future research directions beyond the current study.

 - Suggestion: Conclude with a statement about how this research could pave the way for larger clinical trials or influence treatment protocols.

Totally:

There are minor grammatical issues that could be polished for better readability.

Comments on the Quality of English Language

There are minor grammatical issues that could be polished for better readability.

Author Response

We thank the reviewer for his/her comments, which have largely strengthened our manuscript.

Abstract

  1. The term "underestimated" may not be the most appropriate here. Consider using "underexplored" or "less studied" to better convey the idea that iPolyP has not been thoroughly investigated in this context.

  1. We thank the reviewer. We have changed the word “underestimated” with “less studied” in the abstract paragraph – page 2, line 35.

  1. While you mention that iPolyP is "ubiquitous in all forms of life," it may be beneficial to briefly state its biological significance or functions to provide context for readers unfamiliar with the molecule.

  1. We thank the reviewer for the suggestion. In the first version we included the following sentence: “Made up of hundreds of repeated orthophosphate units, iPolyP is essential for a wide variety of functions in mammalian cells, including the regulation of proliferative signaling pathways. Some evidence has suggested its involvement in carcinogenesis, although more studies need to be pursued.” as a brief description of the molecule. In addition to that we have now added the sentence: “Moreover, iPolyP regulates several homeostatic processes in animals, spanning from energy metabolism to blood coagulation and tissue regeneration.” In the abstract paragraph – page 2, line 39-40.

  1. When mentioning TRPM8, consider briefly explaining its general function or relevance in CRC to enhance understanding.

  1. We thank the reviewer for his/her suggestion. We have added the following sentence: “In addition, iPolyP signaling occurs through the TRPM8 receptor, a well characterized Na+ and Ca2+ ion channel often overexpressed in CRC and linked with poor prognosis, thus promoting CRC cell proliferation.” In the abstract paragraph – page 2, line 43-44.

  1. Specify what these approaches entail (e.g., cell lines used, types of ex-vivo tissues). This will help readers understand the experimental design better.

  1. We thank the reviewer for the suggestion. We added the approached used as follow: “We found that iPolyP is significantly increased in tumor tissues, derived from affected individuals enrolled in this study, compared to the corresponding peritumoral counterparts.” In the abstract paragraph – page 2, line 42. And also: “Pharmacological inhibition of TRPM8 or RNA interference experiments performed in established CRC cell lines, such as Caco-2 and SW620, showed that the involvement of TRPM8 is essential, greater than that of the other two known iPolyP receptors, P2Y1 and RAGE.” In the abstract paragraph – page 2, line 45-47.

  1. Ensure consistent formatting for terms like "in vitro" (use italics) and "ex vivo," as this enhances professionalism and readability.

  1. We thank the reviewer for the correction. We have changed the terms in vitro and ex vivo as suggested throughout the abstract paragraph.

Introduction

  1. Some sentences are long and complex, which can hinder readability. For example, "Individuals bearing CRC display backgrounds that cover a wide range of genetic or epigenetic alterations" could be simplified to "Patients with CRC exhibit a wide range of genetic and epigenetic alterations."

  1. We thank the reviewer for the suggestion. We have changed the mentioned sentence as advised. In the introduction paragraph - page 3, line 58-59.

  1. Improve transitions between sentences and paragraphs to enhance flow. For instance, after discussing genetic mutations, a clearer transition to lifestyle factors would help maintain coherence.

  1. We thank the reviewer for this comment. We have rephrased the following sentence: “Beside the unfavorable genetic predisposition, CRC development embraces an heterogenicity of factors spanning from lifestyle, to environmental mutagens or, more recently, dysbiotic metabolites [6], although mechanistically, a number of queries remain unaddressed and novel potential candidates need to be disclosed.” In the introduction paragraph – page 3, line 64-67.

  1. Use of "belong": The phrase "TRPM8 belongs to the transient receptor potential (TRP) cation channel superfamily" should be corrected to "TRPM8 belongs to the transient receptor potential (TRP) cation channel superfamily."

  1. We thank the reviewer; however, the proposed sentence is identical to the original version.

  1. "iPolyP architecture makes it an obvious participant": The term "obvious" is subjective. Consider using "suggests that iPolyP plays a significant role in energy metabolism."

  1. We thank the reviewer and changed the sentence as follow: “iPolyP architecture suggests its role in energy metabolism;”. In the introduction paragraph – page 3, line 78-79.

  1. When introducing acronyms like RAGE and TRPM8, consider providing a brief description or function to help readers unfamiliar with these terms.

  1. We thank the reviewer and added the description as reported here: “Three binding receptors have been identified for iPolyP, namely: advanced glycosylation end-product specific receptor (RAGE), purinergic receptor P2Y1 and transient receptor potential cation channel subfamily M (melastatin) member 8 (TRPM8) [25,26], displaying entirely different transduction pathways, although all three have been linked to CRC progression and development. In particular, RAGE receptor, known to bind the so-called advanced glycation endproducts (AGE), signals to nuclear factor kappa B (NF-κB) and controls the expression of several genes involved in inflammation, which might foster CRC development [27]. Preliminary evidences lighted up a novel pathway involving iPolyP/RAGE receptor linked to Wingless-related integration site (Wnt)/β-catenin signaling axis, of crucial importance for the regulation of major pathophysiological processes in tumor cells [28]. The purinergic receptor P2Y1, which binds the extracellular ATP, results markedly overexpressed in CRC in respect to the normal counterpart [29], although mechanistic studies need to be performed. Parallelly, recent literature has reported an overexpression of TRPM8 receptor in CRC specimens, which correlates with poorer survival [30]. TRPM8 belong to the transient receptor potential (TRP) cation channel superfamily, subfamily melastatin (M), member 8 (TRPM8), also known as the cold and menthol receptor 1 (CMR1) [31]. However, besides slight evidences about iPolyP-mediated cell proliferation, not many studies have yet reported associations between iPolyP, its receptor and cancer [32], perhaps due to difficulties in iPolyP quantification and lack of comparative data for neoplastic and corresponding normal counterpart.” In the introduction paragraph – page 3, line 83-97.

  1. Mechanisms: The introduction mentions that "mechanistically, a number of queries remain unaddressed." It would be beneficial to specify some of these queries to emphasize the significance of your research.

  1. We thank the reviewer for arising this point. We described some of the challenging queries on the molecular mechanism underpinning CRC onset in this new paragraph: “The complexity of CRC relies on the existence of several different molecular subtypes, characterized by apparently unrelated pathways of development. The major issues are, in-fact, represented by the presence of multiple starting and diverging points during the steps from polyps, adenomas to adenocarcinomas [7]. Gaps in the current understanding of CRC onset include, for instance, the molecular lifestyle and environmental drivers of mutations that cause polyp formation or what background determines the switch to a malignant phenotype. Not least, what make a CRC subtype more responsive to specific therapies than others [8].” In the introduction paragraph – page 3, line 67-72.

  1. Relevance of Citations: Some references (e.g., [15-20]) are broad. Consider specifying key studies or findings related to iPolyP's role in inflammation or tumorigenesis for better context.

  1. We thank the reviewer for this suggestion. iPolyP is only recently appreciated molecule in pathophysiology and literature in the field is still at its early stages. The articles mentioned in the reference list represent the only literature available in our knowledge.

  1. While the introduction identifies a gap in knowledge regarding iPolyP's role in cancer, it could more explicitly state the specific aims or hypotheses of your study at the end of the introduction. This would provide a clearer roadmap for readers.

  1. We thank the reviewer for this suggestion. We added these sentences as reported here: “Aim of our study was firstly to investigate whether iPolyP level were enhanced in tumoral CRC tissue, compared with non-neoplastic counterpart of the same subject. Additionally, we wanted to rule out its role in the development and progression of the disease, as well as the receptor involved in iPolyP signaling, through in vitro and ex vivo strategies.” In the introduction paragraph – page 3-4, line 97-100.

  1. "can spontaneously arise from non-living matter": This statement could be misleading without context. Specify how this occurs or provide a citation to support this claim.

  1. We thank the reviewer for pointing this out. We corrected the sentence as follow: “can be enzymatically produced by living organisms, including bacteria and eukaryotes, or result as product of inorganic catalysis [10].” In the introduction paragraph – page 4, line 74-75.

  1. The final sentences should succinctly summarize the study's objectives. For example, "In this study, we aimed to quantify iPolyP levels in CRC tissues and investigate the role of TRPM8 as a mediator of iPolyP signaling."

  1. We thank the reviewer for this suggestion. We added the aim of our study, as reported in query 13.

Method:

  1. While you mention that patients were candidates for surgical treatment, it would be beneficial to include specific inclusion and exclusion criteria to provide context on how patients were selected for the study. 16. We thank the reviewer for pointing this out. We added these informations as reported here: “Inclusion criteria including patients with a confirmed diagnosis by colonoscopy, biopsy, or imaging studies, in whom surgery was considered beneficial. Patients with a grade of at least 2, i.e., cancer cells look more abnormal, were also considered. The patient to be undergoing surgery also had good health conditions that were good enough to undergo surgery. This includes having a reasonable performance status and no serious comorbidities that would significantly increase the risk of develop complications post-surgery. However, the patient must be willing and able to consent to surgery after being informed of the risks, benefits, and potential outcomes.” This inclusion criteria were added in the paper resubmitted in the Materials and Methods section, Patients’ samples paragraph, page 4, line 115-121.
  2. While you mention studying samples from 50 patients, it would be helpful to provide justification for this sample size. Was it based on power analysis or previous studies? 17. We thank the reviewer for this question. Based on the lack of other published articles in the literature based on polyphosphates and colon cancer, calculating the sample size a priori, or hypothesized a difference between groups was impossible. Therefore, to demonstrate the statistical power of our study a posteriori, we used the effect size on our data. The calculated power was approximately 84% and 10 patients per group would have been sufficient (two-sided and alpha = 0.05). Therefore, this article was statistically correct to demonstrate the difference in polyphosphates concentration between the two independent groups.
  3. Include information on how samples were stored (e.g., temperature, duration) before analysis to ensure reproducibility. 18. We thank the reviewer for pointing this out. We added these informations as reported here: “Samples collected in the operating room are temporarily stored in HypoThermosol FRS (for human cell and tissue preservation - Biolife solutions), sectioned, passed through liquid nitrogen and stored dry in - 80°C within 3 hours.” We added this information in the Materials and Methods section, Patients’ samples paragraph, page 4, line 121-123.
  4. Ensure that terms like "Peritumoral" and "Tumoral" are consistently capitalized throughout the section. We thank the reviewer. We have capitalized Peritumoral and Tumoral throughout the section.
  5. It may be helpful to briefly justify the chosen concentrations of iPolyP and AMTB based on previous studies or preliminary experiments. 20. We thank the reviewer for giving us the opportunity to justify the used concentration. For the iPolyP concentration we performed preliminary titration experiments based on cell proliferation. As reported in the histogram below (see the attached file), 500 nM iPolyP seemed the lowest concentration with effects on proliferation.

    10 mM AMTB was reported in several published articles, with in vitro and in vivo approaches [1-4].

  6. All figures need to be revised to clearly show scale bars.  In addition, the quality of microscopic images needs to be improved. 21. We thank the reviewer for the suggestion. We have improved the quality of the micrographs and revised the scale bar (shown in red).

    Discussion

  7. Some sentences are lengthy and complex, making them difficult to follow. For example, the sentence starting with "The CRC elusiveness is primarily due to..." could be broken down for clarity. 22. We thank the reviewer for the suggestion. We broke down the sentence as follow:” CRC remains one of the deadliest human neoplasms due to its complex interplay of factors including advanced age, environmental influences, and genetic alterations [38, 39]. These elements underlie each stage of tumorigenesis, thus limiting the effectiveness of specific therapy [38].” In the discussion session, page 22, line 539-541.
  8. Simplify sentences and use shorter phrases to enhance readability. For instance:

        - "CRC remains one of the deadliest human neoplasms due to its complex interplay of factors including advanced age, environmental influences, and genetic alterations."

    23. We thank the reviewer for the suggestion. We simplified the sentence as reported in the comment #22.
  9. Maintain consistency in terminology and ensure all abbreviations are defined clearly. 24. We thank the reviewer for the suggestion and ensured the consistency of the terminology.
  10. The section mentions "evidence is still minor" regarding iPolyP's role in cancer, which may undermine the significance of your findings. 25. We thank the reviewer for the suggestion. We have removed the sentence “although the evidence is still minor, perhaps due to the difficulty in visualizing this macromolecule in experimental conditions.” We emphasized the novelty of our findings in the discussion session, page 22, line 547-556.
  11. The transition from discussing iPolyP to its sources (bacterial vs. human) feels abrupt. More context on why this differentiation is important could improve flow.

    - Suggestion: Introduce a transitional sentence that links the importance of understanding the source of iPolyP to its potential therapeutic implications in CRC. 

    26. We thank the reviewer for the suggestion. We added the following sentence: “Understanding the biological source of iPolyP in the context of CRC is a fundamental question, as it could help to design a targeted therapeutic strategy.” In the discussion session, page 22, line 549-550.

    Also, we replaced the sentence:” Moreover, it is largely unknown whether iPolyP has peculiar strains within an altered intestinal flora (a condition termed dysbiosis) which contribute most strongly to iPolyP synthesis.” with “Moreover, it is largely unknown whether iPolyP derives from peculiar strains within an altered intestinal flora (a condition termed dysbiosis) which contribute most strongly to iPolyP synthesis.” In the discussion section, page 22, line 552-553.

  12. Lack of Future Implications: While the discussion touches on potential therapeutic strategies, it lacks a clear direction for future research. 

     - Suggestion: Conclude with a paragraph outlining potential next steps in research, such as exploring specific pathways influenced by iPolyP or investigating its role in other cancer types. 

    27. We thank the reviewer for the suggestion. In the original version we discussed about the next line of research ongoing in our laboratory:” Thus, ongoing mass spectrometry experiments in our laboratory aim firstly to elucidate the provenience of iPolyP within CRC context, steering research toward the appropriate in vivo models targeting either the bacterial or human kinase responsible for the inorganic polyphosphate biosynthesis.” However, we added the following sentence to strengthen the direction of future research: “In addition, we will apply these findings to other cancer types and determine the role of iPolyP in different experimental settings. This might potentially lead to a classification of iPolyP-sensitive neoplasms, or limit its activity solely to the CRC. Finally, we will consider the suitability of iPolyP as novel, non-invasive, biomarker for an early detection of CRC.” In the discussion section, page 23, line 576-579.

    Conclusion

    1. The phrase "involvement of the iPolyP-TRPM8 axis may offer novel therapeutic options" could be more precise about what these options entail.

     - Suggestion: Clarify what specific therapeutic options are being considered. For example, "The findings regarding the iPolyP-TRPM8 axis suggest potential novel therapeutic strategies, such as targeted therapies that enhance TRPM8 activity."

    28. We thank the reviewer for the suggestion. We changed the sentence as follow: “The findings regarding the iPolyP-TRPM8 axis suggest potential novel therapeutic strategies, such as targeted therapies that minimize TRPM8 activity." In the conclusion section, page 23, line 596-597.

29. The statement about in-vivo experimental models being under study lacks detail. It would strengthen the discussion to mention preliminary results or specific methodologies. - Suggestion: Include a brief overview of the experimental design or any preliminary findings to bolster the claim.

29. We thank the reviewer for the suggestion. We included our experimental strategy in the sentence reported here: “In vivo experimental CRC models, involving TRPM8 deficient mice or mice treated with TRPM8 inhibitor, to support our conclusions are currently under study.” In the conclusion section, page 23, line 597-599.

30. The sentence describing early diagnosis methods could benefit from more detail regarding their limitations. - Suggestion: Expand on the invasive nature of current biopsy methods and why they pose challenges for early diagnosis, such as patient discomfort or delayed results.

30. We thank the reviewer for the suggestion. We reported the following sentence: “However, this method poses several challenges and limitations, such as invasiveness, limited accessibility (for tumor located nearby inaccessible areas), resource intensive, since it requires specialized skills, as well as temporal snapshot, which provide a picture of the tumor at a single point in time and might not represents the full heterogeneity of the tumor or catch the dynamic changes it undergoes during treatment.” In the conclusion section, page 24, line 601-605.

31. The aim to test iPolyP as a diagnostic and prognostic biomarker could be more specific about what aspects will be evaluated. - Suggestion: Specify what parameters will be assessed in the study, such as sensitivity, specificity, or correlation with disease stage.

31. We thank the reviewer for the suggestion. We added the following sentence. “Bona fide non-invasive biomarkers should possess high sensitivity property, quantifiable within fluids, including, serum, sweat, urine, saliva or fecal sample. Moreover, it can reflect the body’s responses to tumor progression and/or therapy. Thus, our aim is to test the efficacy of iPolyP as a novel diagnostic and prognostic biomarker to be detected through liquid biopsy or by screening stool samples of CRC-affected subjects.” In the conclusion section, page 24, line 607-610.

32. The transition from discussing therapeutic options to diagnostic methods feels abrupt. - Suggestion: Use transitional phrases to connect these ideas more smoothly. For example, "In addition to potential therapeutic applications, our research also aims to explore..."

32. We thank the reviewer for the suggestion. We added the following sentence.” In addition to potential therapeutic applications, our research also aims to explore whether iPolyP detection may represent a novel approach for the screening of the CRC.” In the conclusion section, page 24, line 599-600.

33. Lack of Forward-Looking Statements: The discussion could benefit from mentioning potential future research directions beyond the current study. - Suggestion: Conclude with a statement about how this research could pave the way for larger clinical trials or influence treatment protocols.

33. We thank the reviewer for the suggestion. We added the following sentence. “Our aim is therefore to test the efficacy of iPolyP as a novel diagnostic and prognostic biomarker to be detected through liquid biopsy or by screening stool samples of CRC-affected subjects. Thus, if successful, this research could pave the way for the development of novel, non-invasive, detection methods, that can offer long-term benefits and improve healthcare, patients' lives with significant social impact.” In the conclusion section, page 24, line 609-612.

References:

  1. Fujino, T. Transient Receptor Potential Melastatin 8, a sensor of cold temperatures mediates expression of cyclin-dependent kinase inhibitor, p21/Cip1, a regulator of epidermal cell proliferation.Toxicol. Sci. 2022; 47, 117-123.
  2. Cao, S.; Li, Q.; Hou, J.; Li, Z.; Cao, X.; Liu, X.; Qin, B. Intrathecal TRPM8 blocking attenuates cold hyperalgesia via PKC and NF-κB signaling in the dorsal root ganglion of rats with neuropathic pain. J. Pain Res. 2019; 12: 1287–1296.
  3. Mueller-Tribbensee, S.M.; Karna, M.; Khalil, M.; Neurath, M.F.; Reeh, P.W.; Engel, M.A. Differential Contribution of TRPA1, TRPV4 and TRPM8 to Colonic Nociception in Mice. PLoS One.2015; 10.
  1. Lin, A.-H.; Liu, M.-H., Ko, H.-K.B.; Perng, D-W.; Lee, T-S.; Kou1, Y.R. Inflammatory Effects of Menthol vs. Non-menthol Cigarette Smoke Extract on Human Lung Epithelial Cells: A Double-Hit on TRPM8 by Reactive Oxygen Species and Menthol. Front Physiol. 2017; 8: 263.

Reviewer 2 Report

Comments and Suggestions for Authors

The manuscript describes the inorganic polyphosphate promotes colorectal cancer growth via TRPM8 receptor signaling pathway. The paper is interesting and may be suitable for the Journal, but first several improvements must be performed:

1.      Inorganic polyphosphate (iPolyP) is essential for a wide variety of functions in mammalian cells. Authors found that iPolyP is significantly increased in colorectal tumor tissues compared to the corresponding peritumoral counterparts. Authors should discuss what reasons cause iPolyP is significantly increased in colorectal tumor tissues.

2.      Whether other tumor tissues also increase iPolyP, authors must compare and explore. Only then can we determine that the high P can be used as a novel pivotal biomarker linked with colorectal cancer growth.

3.      The authors only analyzed PCNA and did not analyze the cell cycle in detail. I suggest that the authors analyze related proteins that regulate the cell cycle, such as cyclin D, cyclin E, cdk2, cdk4, cdk6 and ppRb etc. In this way, the role of iPolyP in promoting the proliferation of colorectal cancer can be further explored.

4.      The research is interesting, but the conclusions are too general. The Authors should write significant limitations of the study.

Author Response

We thank the reviewer for his/her comments, which helped us to improve the quality of this manuscript.

  1. Inorganic polyphosphate (iPolyP) is essential for a wide variety of functions in mammalian cells. Authors found that iPolyP is significantly increased in colorectal tumor tissues compared to the corresponding peritumoral counterparts. Authors should discuss what reasons cause iPolyP is significantly increased in colorectal tumor tissues. 1. We thank the reviewer for the suggestion which gave us the opportunity to explain better this crucial point. So far, there are several different sources of iPolyP in mammals. It can derive from active platelets, or, in case of neoplasm, from cancer cells; moreover, it can originate from intestinal microbiota as well. Hence, the reasons underpinning high level of iPolyP in CRC samples could lie in the altered cancer cells metabolism, or in an over-production of platelet’s dense granules, or even in a dysbiotic microbiota with specific strains-producing inorganic polyphosphate. We have, therefore, added the following sentence as reported here: “Hence, although our findings point out, for the first time, the presence of iPolyP within tumoral samples of CRC-affected individuals, a significant limitation in the study is the lack of the informations regarding the origin, as it can be either from active platelets infiltrating the tumor, cancer cells or dysbiotic intestinal microbiota. This knowledge will allow the development of a targeted therapy. In the discussion paragraph, page 23, line 570-573.
  2. Whether other tumor tissues also increase iPolyP, authors must compare and explore. Only then can we determine that the high P can be used as a novel pivotal biomarker linked with colorectal cancer growth. 2. We thank the reviewer for the suggestion. We are actively planning to perform similar analysis on other tumor tissues, and, as indicated from reviewer #1, we added the following sentence: “In addition, we will apply these findings to other cancer types and determine the role of iPolyP in different experimental settings. This might potentially lead to a classification of iPolyP-sensitive neoplasms, or limit its activity solely to the CRC.” In the discussion section, page 22, line 576-578.
  3. The authors only analyzed PCNA and did not analyze the cell cycle in detail. I suggest that the authors analyze related proteins that regulate the cell cycle, such as cyclin D, cyclin E, cdk2, cdk4, cdk6 and ppRb etc. In this way, the role of iPolyP in promoting the proliferation of colorectal cancer can be further explored.  

    3. We thank the reviewer for the suggestion. We extended the transcriptional analysis to the following genes, as suggested. Reported in the attached file, is the relative amount of mRNA upon treatment with iPolyP in Caco-2 cell line. Moreover, we included these data in a new Supplementary Figure S8 and rephrase the following sentence: “Although we noticed a considerable increase, in terms of number of copies of transcripts related to genes like ccnd1 (cyclin D1), ccna1 (cyclin A1), ccne1 (cyclin E1), and cyclin-dependent kinases-2, -4 and -6 (cdk2, cdk4 and cdk6, respectively), mainly involved in the G1 and S/G2 phases, a significant signal amplification was observed for the ccnb1 transcript, which codes for the mitotic protein cyclin B, a master regulator of the G2/M phase (Figure 6A and Supplementary Figure S6A).” In the results section, page 21, line 498-502.

    1. The research is interesting, but the conclusions are too general. The Authors should write significant limitations of the study.

    4. We thank the reviewer for the suggestion. We added the limitation of the study within the same sentence reported in query #2: “Hence, although our findings point out, for the first time, the presence of iPolyP within tumoral samples of CRC-affected individuals, a significant limitation in the study is the lack of the informations regarding the origin, as it can be either from active platelets infiltrating the tumor, cancer cells or dysbiotic intestinal microbiota. This knowledge will allow the development of a targeted and specific therapy.” In the discussion section, page 23, line 570-573.

Round 2

Reviewer 1 Report

Comments and Suggestions for Authors

Dear Editor,

I have completed the review of the manuscript titled "Inorganic polyphosphate promotes colorectal cancer growth via TRPM8

receptor signaling pathway", and I am pleased to recommend its acceptance for publication in Cancers Journal.

Best,

Reviewer 2 Report

Comments and Suggestions for Authors

Authors had answer my questions and revised the manuscript according my suggestions. I suggest the manuscript can be accepted and published on the journal.